# Parkin Promotes Airway Inflammatory Response to Interferon Gamma

**DOI:** 10.3390/biomedicines11102850

**Published:** 2023-10-20

**Authors:** Kris Genelyn Dimasuay, Niccolette Schaunaman, Bruce Berg, Taylor Nichols, Hong Wei Chu

**Affiliations:** Department of Medicine, National Jewish Health, 1400 Jackson Street, Denver, CO 80206, USA

**Keywords:** Parkin, IFN-γ, Thap11, lung, inflammation

## Abstract

Purpose: Increased type 2 interferon (i.e., IFN-γ) signaling has been shown to be involved in airway inflammation in a subset of asthma patients who often show high levels of airway neutrophilic inflammation and poor response to corticosteroid treatment. How IFN-γ mediates airway inflammation in a mitochondrial dysfunction setting (e.g., Parkin up-regulation) remains poorly understood. The goal of this study was to determine the role of Parkin, an E3 ubiquitin ligase, in IFN-γ-mediated airway inflammation and the regulation of Parkin by IFN-γ. Methods: A mouse model of IFN-γ treatment in wild-type and Parkin knockout mice, and cultured human primary airway epithelial cells with or without Parkin gene deficiency were used. Results: Parkin was found to be necessary for the production of neutrophil chemokines (i.e., LIX and IL-8) and airway neutrophilic inflammation following IFN-γ treatment. Mechanistically, Parkin was induced by IFN-γ treatment both in vivo and in vitro, which was associated with less expression of a Parkin transcriptional repressor Thap11. Overexpression of Thap11 inhibited Parkin expression in IFN-γ-stimulated airway epithelial cells. Conclusions: Our data suggest a novel mechanism by which IFN-γ induces airway neutrophilic inflammation through the Thap11/Parkin axis. Inhibition of Parkin expression or activity may provide a new therapeutic target for the treatment of excessive neutrophilic inflammation in an IFN-γ-high environment.

## 1. Background

Asthma is a heterogenous inflammatory airway disease characterized by various endotypes. The most common endotypes are type 2 inflammation-high and type 2 inflammation-low. Type 2-high asthma is featured by increased type 2 inflammation including higher levels of IL-4, IL-5, and IL-13 release and infiltration of eosinophils in the airways [1]. In contrast, airways of type 2-low asthma demonstrate neutrophilic and pauci-granulocytic airway infiltrates that are associated with cytokines such as interferon gamma (IFN-γ), IL-6 and IL-17 [2]. Currently, there are limited therapeutic options for type 2-low asthma compared to type 2-high asthma. In a study by Ricciardolo et al. [3], high levels of IFN-γ expression were found in type 2-low asthma subjects. Additionally, increased IFN-γ in bronchoalveolar lavage (BAL) fluid was associated with neutrophilic inflammation in severe asthma patients [4]. This was further supported by a recent study demonstrating the association of increased IFN-γ expression in airway samples from a cohort of asthma patients with more severe disease despite corticosteroid use, one of the major anti-inflammatory approaches to treat asthma [5]. The mechanisms of increased IFN-γ expression are still under investigation, although several major factors such as viral and bacterial infections have been proposed [6,7]. While corticosteroids such as dexamethasone have been shown to inhibit IFN-γ signaling in vitro [8], they did not reduce IFN-γ levels in a mouse model of asthma [9]. The fact that severe asthma patients presented higher levels of IFN-γ even with treatment of high doses of corticosteroids [4] suggests the urgent need to further understand how IFN-γ-mediated inflammation is regulated.

Parkin, an E3 ubiquitin ligase, is encoded by *PARK2* gene and expressed by various tissues (e.g., brain, kidney, lung) and different types of cells including neural cells and non-neural cells such as lung epithelial cells. Parkin protein consists of a N-terminal ubiquitin-like domain, the linker region and the C-terminal RING-box [10,11,12]. As an intracellular protein, the primary function of Parkin is related to its E3 ubiquitin ligase activity, a key component in 26S proteasome-mediated protein degradation. Recently, Parkin has been shown to exert non-classical/novel functions such as a pro-inflammatory role [13]. We have previously investigated the regulation of airway type 2 inflammation by Parkin [14]. Specifically, Parkin promoted type 2 airway inflammation in a human airway epithelial cell air-liquid interface culture system, and in mouse models of IL-13 treatment and allergen challenges. A recent study [15] suggests that Parkin impairs host antiviral immunity by inhibiting the antiviral inflammation, but not affecting the type 1 interferon response. Whether Parkin regulates airway type 2 interferon (i.e., IFN-γ) response is unknown.

Regulation of Parkin expression or function in the airways has not been well understood. We have demonstrated that Parkin expression in airway epithelium of asthma patients was increased [14]. A previous study utilizing the genome-wide CRISPR screening approach identified transcription factor Thap11 as a major repressor of Parkin expression as Thap11 knockdown was shown to increase Parkin expression and activity [16]. It is unclear whether IFN-γ may regulate Thap11 expression, and subsequently affect Parkin expression. By using the human airway epithelial cell culture system and mouse models, we determined the interplay of IFN-γ, Thap11 and Parkin by testing whether IFN-γ upregulates Parkin by inhibiting Thap11, and whether Parkin in turn amplifies host pro-inflammatory response to IFN-γ.

## 2. Methods

### 2.1. Intranasal Administration of Recombinant IFN-γ Protein in Mice

Wild-type (WT) C57BL/6 and Parkin knockout (PKO) mice on C57BL/6 background were purchased from the Jackson Laboratory (Bar Harbor, ME, USA), and housed at the National Jewish Health (NJH) Biological Resource Center under pathogen-free housing conditions. All mouse procedures were approved by the Institutional Animal Care and Use Committee (protocol #AS2792-03-23) at National Jewish Health. 

To induce lung inflammation in mice, WT and PKO mice were intranasally inoculated with recombinant mouse IFN-γ (PeproTech, Cranbury, NJ, USA) at 25 ng/mouse prepared in 50 µL of phosphate-buffered saline (PBS) containing 0.1% bovine serum albumin (BSA) or 50 µL of PBS containing 0.1% BSA (control) once daily for three consecutive days. We used this dose based on our pilot experiment comparing lung neutrophil levels between 10 ng/mouse and 25 ng/mouse, and we found that 25 ng/mouse induced about 15% of neutrophils in bronchoalveolar lavage fluid (BALF), while 10 ng/mL IFN-γ resulted in minimal lung neutrophilic inflammation. Raundhal et al. reported this similar percentage of neutrophils in BALF from human severe asthma subjects with more IFN-γ+ T cells in the airways [4]. 

Mice were sacrificed after 24 h of the last IFN-γ treatment. BAL was performed with 1 mL of sterile saline. BAL cells were used for the leukocyte count and BALF was analyzed for pro-inflammatory mediators.

### 2.2. IFN-γ Stimulation in Cultured Primary Human Airway Epithelial Cells

Human tracheobronchial epithelial (HTBE) cells from donors (n = 6) without lung disease or smoking history were used for cell culture. The human study was approved by the National Jewish Health Institutional Review Board (Protocol # HS-3209). HTBE cells were cultured in 12-well cell culture plates (10^5^ cells/well) under the submerged condition in the presence or absence of IFN-γ (5 ng/mL) for 24 h to measure Thap11 and Parkin protein expression. Submerged cell culture was performed because IFN-γ was found to similarly increase airway epithelial Parkin expression in our submerged and air-liquid interface (ALI) culture experiments. In our preliminary experiment, we compared the regulation of Parkin by IFN-γ in submerged and ALI culture. We found that Parkin was similarly up-regulated by IFN-γ in both cell culture models. Thus, we utilized submerged culture model to determine the regulation of Parkin and Thap11 by IFN-γ. We also performed a time course (24 and 48 h) and dose response (5 and 10 ng/mL) study of IFN-γ stimulation in human airway epithelial cells (n = 3 subjects). We found a consistent decrease in Thap11 by 5 ng/mL of IFN-γ at 24 h. In most of the previous human lung epithelial cell culture publications, IFN-γ concentrations ranged from 1 to 10 ng/mL [17,18]. IFN-γ dose (5 ng/mL) used in our study falls within the IFN-γ dosing range of previous work.

### 2.3. Thap11 Overexpression in Primary Human Airway Epithelial Cells

To determine the role of Thap11 in Parkin expression in the absence or presence of IFN-γ, Thap11 overexpression was performed in HTBE cells from a donor without lung disease and smoking history. Lentivirus system was utilized to overexpress Thap11 using reagents from GeneCopoeia, Inc. (Rockville, MD, USA). Thap11 open reading frame (ORF) or scrambled control sequence was cloned into a lentiviral vector (pReceiver-Lv205) and subsequently packaged in 293FT cells using the Lenti-Pac HIV Expression Packaging Kit. After 48 h, the supernatant containing the viral particles was harvested. HTBE cells grown in 24-well plates under submerged condition were transduced with 70% viral particle supernatant plus 30% culture medium or 100% viral particle supernatant for 72 h before treatment with IFN-γ (10 ng/mL) [17]. After 24 h, epithelial supernatant was collected for ELISA while cell lysates were harvested in RIPA for Western blot. 

### 2.4. Parkin Knockout (PKO) in Primary HTBE Cells

Control and PKO HTBE cells from a donor without lung disease and smoking history were generated using CRISPR-Cas9 as detailed in our previous publication [14]. Briefly, a single guide RNA (sgRNA, 5′ AGTCTAAGCAAATCACGTGG 3′) was designed to target exon 7 of human Parkin in HTBE cells. For the control CRISPR, a scrambled sgRNA was used. The transduced HTBE cells were expanded in collagen-coated 60 mm tissue culture dishes containing BronchiaLife epithelial medium (Lifeline Cell Technology, Frederick, MD, USA), and then plated onto collagen-coated 12-well transwells (Corning Inc., Corning, NY, USA). Briefly, cells on the transwells were under submerged culture for 7–10 days to form a monolayer, and then cultured at ALI for 21 days in Pneumacult ALI medium (Stemcell Technologies, Vancouver, BC, Canada) to promote mucociliary differentiation as previously described [14]. At day 21, cells were stimulated with recombinant human IFN-γ (5 ng/mL) or 0.1% BSA (control solution for preparing IFN-γ) for three days. Basolateral supernatants were collected for ELISA while cell lysates were harvested in RIPA for Western blot analysis. 

### 2.5. Western Blot

Cell lysates (15 µg of total protein) were separated by SDS-PAGE, transferred onto PVDF membranes using a semidry transfer protocol, blocked with blocking buffer, and incubated with antibodies against Parkin, Thap11, or β-actin (1:500, Santa Cruz Biotechnology, Dallas, TX, USA). After washes in PBS with 0.1% Tween-20, the membranes were incubated with the appropriate horseradish peroxidase (HRP)-linked secondary antibodies (1:3000; EMD Millipore, Burlington, MA, USA) and developed using a Fotodyne imaging system (Fotodyne, Inc., Hartland, WI, USA). Densitometry was performed to quantify protein expression levels using the National Institutes of Health’s ImageJ software (Version 1.53t).

### 2.6. ELISA

Human IL-8 (catalog #: DY208) and mouse LIX/CXCL5 (catalog #: DY443) were measured using Duoset ELISA kits from R&D systems (Minneapolis, MN, USA).

### 2.7. Statistical Analyses 

Data were analyzed using Graph Pad Prism software (Version 9.5.1). For parametric data with a normal distribution, a paired Student’s *t*-test was performed for two-group comparisons or two-way ANOVA followed by Tukey’s multiple comparison test. For non-parametric data, comparisons were carried out using the Mann–Whitney test for two group comparisons or using the Kruskal–Wallis test for multiple group comparisons. A *p* value of <0.05 was considered to be statistically significant. 

## 3. Results

### 3.1. Parkin Was Essential to Mouse Lung Neutrophilic Inflammation Induced by IFN-γ

To determine the in vivo function of Parkin in lung neutrophilic inflammation following IFN-γ treatment, wild-type (WT) and Parkin knockout (PKO) mice were treated with IFN-γ. WT mice treated with IFN-γ increased the number and percentage of neutrophils (Figure 1a) as well as neutrophilic chemoattractant LIX in bronchoalveolar lavage fluid (BALF) (Figure 1b). PKO mice were also able to significantly increase LIX after IFN-γ treatment, but LIX levels in PKO mice were lower than those in WT mice treated with IFN-γ. The other major type of inflammatory cells in BAL is macrophages. While the number of macrophages trended to be lower in PKO mice than the wild-type mice after IFN-γ treatment, the % of macrophages trended to be higher (Figure 1c). These findings suggest that Parkin enhances neutrophilic inflammation after IFN-γ challenge in vivo. As expected, Parkin was deficient in the lungs of PKO mice (Figure 1d). Notably, IFN-γ significantly induced Parkin expression in WT mouse lung tissues. 

### 3.2. IFN-γ Increased Parkin, but Inhibited Thap11 Expression in Primary Human Airway Epithelial Cells 

Next, we sought to determine whether IFN-γ increases Parkin expression in human airway epithelial cells, which is associated with a reduction in Thap11, a transcriptional repressor of Parkin [16]. In submerged culture of primary human airway epithelial cells, IFN-γ significantly increased Parkin expression (Figure 2a), which further supported our mouse model data. Importantly, increased Parkin expression by IFN-γ was accompanied by decreased Thap11 expression (Figure 2b), suggesting a negative relationship between Thap11 and Parkin under IFN-γ stimulation. 

### 3.3. Overexpression of Thap11 Decreased Parkin Expression in Primary Human Airway Epithelial Cells Treated with IFN-γ

Knockdown of Thap11 in the HEK293 cell line was shown to increase Parkin mRNA and protein expression in a previous study [16]. Whether Thap11 regulates Parkin expression in human primary airway epithelial cells has not been investigated. To test if Thap11 regulates Parkin, we overexpressed Thap11 in HTBE cells with two different doses of lentiviral particles containing the Thap11 open reading frame (ORF) plasmid. Cells transduced with 100% lentivirus particles expressed Thap11 protein that was about five-fold higher than the control cells (Figure 3a). Although cells transduced with 70% viral particles showed higher Thap11 expression, it was not significantly higher than the control cells. We measured IL-8 to determine if the lentivirus particles induced a pro-inflammatory response. Notably, IL-8 levels were similar between cells transduced with 70% or 100% viral particles and control cells (Figure 3b), suggesting that the lentiviral particles did not induce the pro-inflammatory response. We proceeded to stimulate the cells receiving 100% viral particles with IFN-γ. Enhancing Thap11 expression in HTBE cells stimulated with IFN-γ decreased Parkin expression as well as IL-8 levels compared to the scrambled control cells (Figure 3c,d). This suggests that restoring Thap11 expression in IFN-γ-stimulated cells decreased Parkin expression and the pro-inflammatory response. 

### 3.4. Parkin Is Essential to Maintain Human Airway Epithelial Pro-Inflammatory Response to IFN-γ

Primary HTBE cells with Parkin knockout using the CRISPR-Cas9 approach were utilized to test if Parkin induction via IFN-γ contributes to the production of pro-neutrophilic molecules. Similar to the mouse lung and submerged HTBE culture data, IFN-γ increased Parkin protein expression in scrambled control CRISPR cells (Figure 4a). We confirmed the loss of Parkin protein expression in the absence and particularly in the presence of IFN-γ stimulation. The pro-neutrophilic chemokine IL-8 was increased in IFN-γ-stimulated scrambled control cells compared to untreated cells (Figure 4b). Importantly, Parkin deficiency decreased IL-8 production after IFN-γ stimulation, suggesting that Parkin may maintain or promote a pro-neutrophilic response to IFN-γ.

## 4. Discussion

Whether Parkin regulates type 2-low (e.g., IFN-γ-high) lung inflammatory responses has not been reported. Here, for the very first time, we demonstrated that Parkin promotes IFN-γ-mediated lung neutrophilic inflammation. Moreover, we discovered that IFN-γ serves as a positive feedback loop mechanism to increase Parkin expression. 

We have reported increased Parkin expression in asthmatic airway epithelium [14]. However, the role of Parkin in lung inflammation has not been well understood. An early mouse model study suggested that Parkin regulates lipopolysaccharide (LPS)-induced pro-inflammatory responses as Parkin-deficient mice demonstrated less severe lung inflammation (e.g., neutrophils) as compared to the wild-type mice [19]. Our data generated in Parkin-deficient mice treated with IFN-γ further suggest the role of Parkin in promoting lung neutrophilic inflammation. How Parkin promotes or maintains the pro-inflammatory response to IFN-γ remains to be explored. IFN-γ-mediated effects have been shown to be associated with activation of STAT1 and NF-κB [20,21]. A previous study showed that a RBR E3 ubiquitin ligase Natural Killer Lytic-Associated Molecule (NKLAM) may interact with STAT1 to maintain STAT1-mediated transcriptional activity in macrophages [22]. Parkin has been functionally considered as a RBR E3 ligase, but recent studies suggest that it may also act as RING-between-RING and HECT E3 ligases [23]. It remains to be determined if Parkin promotes IFN-γ-mediated inflammatory response through increasing STAT1-induced gene expression. Additionally, Parkin was shown to up-regulate NF-κB activation [24], which may be another potential mechanism underlying the pro-inflammatory effect of Parkin. 

We not only found that Parkin regulates IFN-γ-mediated inflammatory response, but also discovered that Parkin was up-regulated by IFN-γ stimulation. Parkin expression can be regulated at multiple (e.g., genetic, epigenetic, transcriptional) levels. While transcription factor activating transcription factor 4 (ATF4) was shown to up-regulate Parkin under ER stress [25], Thap11 may serve as a transcriptional repressor of Parkin [16]. Our data in airway epithelial culture demonstrated that IFN-γ reduced Thap11, which was coupled with increased Parkin expression. In IFN-γ-stimulated airway epithelial cells, Thap11 overexpression reduced Parkin expression. Together, our data suggest that IFN-γ increases Parkin expression in part through reducing the expression of Thap11. At the moment, it is unclear whether IFN-γ may also regulate E3 ligase activity of Parkin, but this warrants further investigation in order to fully understand the mutual regulation of IFN-γ and Parkin. 

We realize several limitations to this study. First, we utilized an acute mouse model of IFN-γ treatment to study the role of Parkin in the inflammatory process to mimic the inflammatory process in asthma patients with acute exacerbations which pose a significant healthcare challenge and could contribute to fatal asthma. Future studies are necessary to determine the role of Parkin in chronic type 2-low inflammation induced by IFN-γ. Second, while we provided data supporting the role of the Thap11/Parkin axis in IFN-γ-mediated airway neutrophilic inflammation, future studies are warranted to further determine how IFN-γ regulates Parkin and Thap11 expression, and whether Parkin affects IFN-γ signaling (e.g., STAT1 activation). Third, a previous study has shown the ability of IFN-γ to enhance ATF4 mRNA expression under ER stress in pancreatic beta cells [26]. Whether IFN-γ utilizes other transcription factors such as ATF4 to up-regulate Parkin in human airway epithelial cells could be further studied. Fourth, we established Thap11 overexpressing and Parkin knockout airway epithelial cell lines only from one human donor. While these cell lines provided clear evidence about the role of Thap11 and Parkin in IFN-γ-mediated inflammatory response, our results may need to be replicated in cells from additional donors. Lastly, we generated gene overexpression (i.e., Thap11) or knockout (i.e., Parkin) from a single donor to demonstrate their biological functions. Although we expect similar results using cells from different donors with gene overexpression or knockout, this needs to be confirmed in future studies.

## 5. Conclusions

Our current study has deepened our understanding about the role of Parkin in type 2-low inflammation (Figure 5). As Parkin dysregulation may involve multiple biological processes such as mitochondrial dysfunction, it is conceivable that unraveling its regulation and functions under physiological and pathological conditions may provide insights into mechanisms of a variety of diseases with elevated IFN-γ or type 1 or type 3 interferons as seen in respiratory viral infections.

## Figures and Tables

**Figure 1 biomedicines-11-02850-f001:**
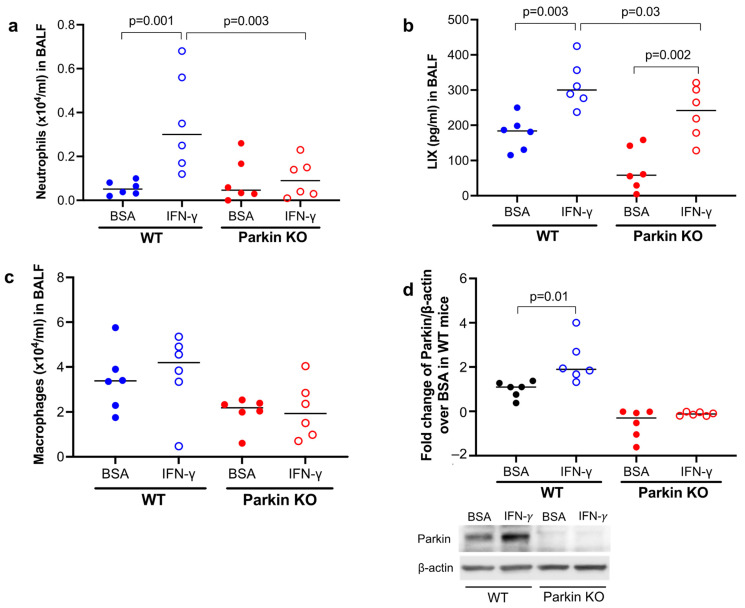
Parkin deficiency reduced IFN-γ-mediated lung inflammatory response in mice. Wild-type (WT) and Parkin knockout (KO) mice were intranasally treated with IFN-γ for 24 h. (**a**) Neutrophils in bronchoalveolar lavage fluid (BALF) of WT and Parkin KO mice treated with IFN-γ and bovine serum albumin (BSA, control for IFN-γ). (**b**) Levels of neutrophil chemokine LIX in BALF. (**c**) Macrophages in BALF of WT and Parkin KO mice. (**d**) Western blot densitometry and images of Parkin showing Parkin deficiency in Parkin KO lung tissue and Parkin up-regulation by IFN-γ in WT mouse lung. Western blot data were obtained from independent samples. N = 6 mice/group. The number of mice for each group was decided based on our previous publication [14] studying the role of Parkin in airway type 2 inflammation where 5 to 7 mice per experimental group were included. The horizontal bars represent medians.

**Figure 2 biomedicines-11-02850-f002:**
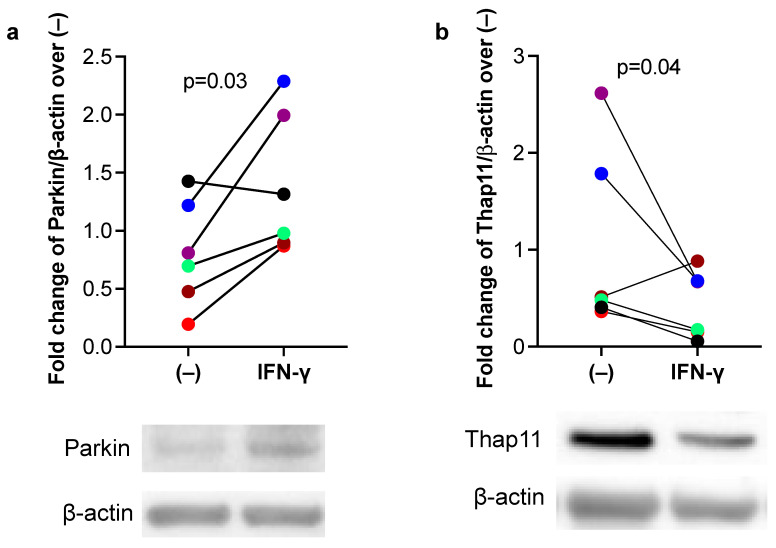
IFN-γ increased Parkin, but decreased Thap11 protein expression in human primary airway epithelial cells. Tracheobronchial epithelial cells from 6 donors without smoking history and lung disease were cultured under the submerged condition with or without recombinant human IFN-γ protein for 24 h. (**a**) Western blot of Parkin and densitometry data. (**b**) Western blot of Thap11 and densitometry data. Different colors represent data obtained from different donors. The horizontal bars represent medians.

**Figure 3 biomedicines-11-02850-f003:**
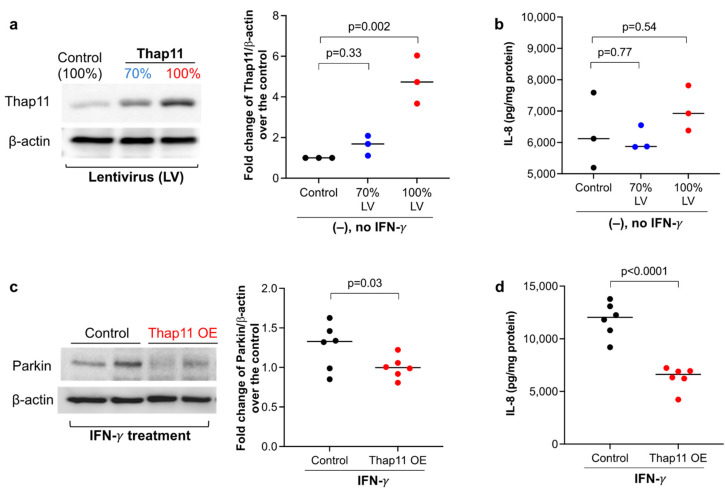
Thap11 overexpression reduced Parkin expression and IL-8 production in primary normal human airway epithelial cells treated with IFN-γ. (**a**) Healthy donor tracheobronchial epithelial cells transduced with Thap11 cDNA encoding lentivirus (LV) increased Thap 11 protein expression in a dose-dependent manner. (**b**) IL-8 levels in supernatants of Thap11-overexpressing epithelial cells without IFN-γ stimulation. (**c**,**d**) In IFN-γ-stimulated epithelial cells, Thap11 overexpression (OE) reduced intracellular Parkin expression and IL-8 levels in supernatant. N = 6 replicates from one donor Thap11 overexpressing cell line. The horizontal bars represent medians.

**Figure 4 biomedicines-11-02850-f004:**
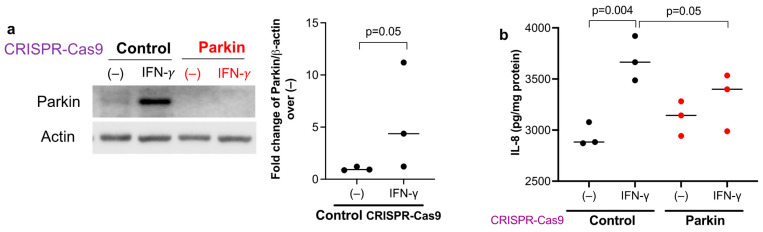
Parkin deficiency in human airway epithelial cells reduced the production of IL-8 induced by IFN-γ. (**a**) Parkin expression was reduced in healthy donor tracheobronchial epithelial cells transduced with Parkin CRISPR construct as compared with the scrambled control CRISPR. The cells were cultured at air-liquid interface for 3 weeks, and then treated with IFN-γ for 72 h. IFN-γ increased Parkin expression in epithelial cells transduced with the scrambled control CRISPR. (**b**) Parkin deficiency reduced the induction of IL-8 by IFN-γ. N = 3 replicates from one donor Parkin deficient cell line. The horizontal bars represent medians.

**Figure 5 biomedicines-11-02850-f005:**
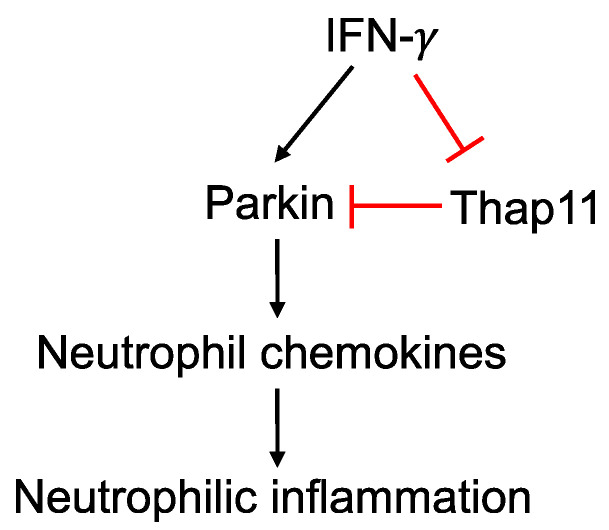
Proposed mechanisms by which Parkin promotes IFN-γ-mediated airway neutrophilic inflammation. While IFN-γ increases Parkin expression, it inhibits the expression of Thap11 which serves as an inhibitor of Parkin expression. Parkin up-regulation enhances the production of neutrophil chemokines, leading to neutrophil recruitment into the lung and neutrophilic inflammation.

## Data Availability

All data generated or analyzed during this study are included in this published article.

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
