# Peer review of "Parkin Promotes Airway Inflammatory Response to Interferon Gamma"

_biomedicines, 2023, doi:10.3390/biomedicines11102850_

Round 1

Reviewer 1 Report

This manuscript demonstrates that Parkin regulates the lung low inflammatory response via thap 11. This is a well conducted and written study. I have a few minor comments 

1. The volume and vehicle of the .1 % BSA solution instillation in the mice need to be included.

2. Given the variability in the protein expressions in different subjects demands inclusion of more 'n".

3. For all cell experiments the number of donors should be included. The details if the experiment was done at ALI or in submerged cells should be included. Some experiments in submerged and some in ALI should be presented with a good rationale.

Author Response

This manuscript demonstrates that Parkin regulates the lung low inflammatory response via thap 11. This is a well conducted and written study. I have a few minor comments.

We thank this reviewer for your time to review our manuscript. Your comments are very helpful to improve our manuscript. The changes we made are marked in red.

  1. The volume and vehicle of the .1 % BSA solution instillation in the mice need to be included.

The volume (50 µl) and vehicle (Phosphate-buffered saline, PBS) of the 0.1 % BSA solution instillation in the mice have now been included in the revised manuscript (page 5, lines 99 and 100).

  1. Given the variability in the protein expressions in different subjects demands inclusion of more 'n".

We agree with the protein data variability in mice and human subjects. For Figures 1 and 2 where mouse model and human primary cell cultures were performed, we had n=6 in each group. As we responded to reviewer 2’s comment, the sample size of 6 was based on our preliminary data and power calculation.  For Figures 3 and 4, some of samples with n=3 replicates from the same human subject, which is due to the fact that the data was generated from the cell line we created for gene overexpression or gene knockout for the mechanistic studies. Unlike the primary cells from individual subjects, the cell lines with the gene manipulation (overexpression or knockout) are intended for the proof-of-concept studies as most previous studies used the cell line created from a single human donor. We have acknowledged this in the Discussion section (page 14, lines 296 to 300): “We generated gene overexpression (i.e., Thap11) or knockout (i.e., Parkin) from a single donor to demonstrate their biological functions. Although we expect similar results using cells from different donors with gene overexpression or knockout, this needs to be confirmed in future studies.”

  1. For all cell experiments the number of donors should be included. The details if the experiment was done at ALI or in submerged cells should be included. Some experiments in submerged and some in ALI should be presented with a good rationale.

We have indicated the number of donors in the figure legends.

We have provided more details for air-liquid interface (ALI) culture (page 7, lines 150 to 156): “The transduced HTBE cells were expanded in collagen-coated 60-mm tissue culture dishes containing BronchiaLife epithelial medium (Lifeline Cell Technology, Frederick, MD), and then plated onto collagen-coated 12-well transwells (Corning Inc., Corning, NY). Briefly, cells on the transwells were under submerged culture for 7–10 days to form a monolayer, and then cultured at air-liquid interface (ALI) for 21 days in Pneumacult ALI medium (Stemcell Technologies, Vancouver, Canada) to promote mucociliary differentiation.”

The rationale for using both submerged and ALI culture has been provided (page 6, lines 119 to 128). “In our preliminary experiment, we compared the regulation of Parkin by IFN-gamma in submerged and air-liquid interface (ALI) culture. We found that Parkin was similarly up-regulated by IFN-gamma in both cell culture models. Thus, we utilized submerged culture model to determine the regulation of Parkin and Thap11 by IFN-gamma”.

Reviewer 2 Report

The manuscript entitled “Parkin promotes airway inflammatory response to interferon gamma” provides insights into the role of Parkin, an E3 ubiquitin ligase, in IFN-gamma-mediated airway inflammation using a mouse model (wild-type and Parkin knockout), and cultured human primary airway epithelial cells. Initially, the authors proved that Parkin was essential for the generation of LIX and IL-8 chemokines and airway neutrophilic inflammation. Specifically, IFN-gamma treatment induced Parkin expression, which was associated with decreased expression of Thap11, a transcriptional repressor of Parkin. Overall, the current study suggested a novel mechanism by which IFN-gamma can induce airway neutrophilic inflammation through the Thap11/Parkin axis. The current findings are interesting.

Comments:     

1) Did the authors consider that studying the role of Parkin in airway inflammation in an acute rodent model only may not give the whole idea of its role in inflammation? Why did the authors not study the role of parkin in a chronic animal model of airway inflammation?

2) Likewise, studying the role of parkin in epithelial cell lines from one human sample may not give adequate insights. Replication of this experiment in epithelial cell lines from other healthy donors would be expected.

3) A variety of factors, including environmental and genetic factors, microbial pathogens, and upregulation of inflammatory mediators, may be involved in the development of airway inflammation, but the detailed causes remain unknown. Based on these backgrounds, the used experimental model in this study (interferon gamma-induced airway inflammation in mice) should be described in detail in the introduction. The inhibitory effects of other agents on this model should be addressed in the introduction.

4) In section 2.2., how were the dose of IFN-gamma (5 ng/ml) and the exposure time (24h) decided by the authors? Authors are advised to address this point and add the answers/proper citations to this section.

5) In the material and methods section, please comment on the adequacy of sample size calculation. How did the authors decide on using the listed number of animals per experimental group? Please, clarify for readers.

6) In Western blotting (section 2.5), the authors should describe in more detail the way Western blotting was conducted. How many micrograms were loaded into the gel, how was the transfer done (wet or semidry), how was band visualization performed; did you use x-ray film or specific imaging equipment? Please, add the answer to the comment to this section.

7) In section 2.6, please add the catalog numbers for the used kits.

8) In the statistical analysis section, did the authors check data normality before proceeding to one-way ANOVA? Authors are advised to address this point and add the answers to the comment in this section.

9) In Figure legends, the authors are advised to clarify in Western blotting whether the data were extracted from independent samples.

10) The full name of abbreviations should be given at the first mention of the abbreviation in the text. For example, BAL in line 85. Please address this issue in the entire manuscript.

Minor editing of the English language is required.

Author Response

The manuscript entitled “Parkin promotes airway inflammatory response to interferon gamma” provides insights into the role of Parkin, an E3 ubiquitin ligase, in IFN-gamma-mediated airway inflammation using a mouse model (wild-type and Parkin knockout), and cultured human primary airway epithelial cells. Initially, the authors proved that Parkin was essential for the generation of LIX and IL-8 chemokines and airway neutrophilic inflammation. Specifically, IFN-gamma treatment induced Parkin expression, which was associated with decreased expression of Thap11, a transcriptional repressor of Parkin. Overall, the current study suggested a novel mechanism by which IFN-gamma can induce airway neutrophilic inflammation through the Thap11/Parkin axis. The current findings are interesting.

We thank this reviewer for your interests in our research findings and your very constructive comments, which are very helpful to further improve our manuscript. The changes we made are marked in red.

Comments:     

1) Did the authors consider that studying the role of Parkin in airway inflammation in an acute rodent model only may not give the whole idea of its role in inflammation? Why did the authors not study the role of parkin in a chronic animal model of airway inflammation?

We first chose the acute treatment model of IFN-gamma treatment to mimic the inflammatory process in asthma patients with acute exacerbations which poses a significant healthcare challenge and could contribute to fatal asthma. We have further clarified this in the Discussion. We will plan to study the role of Parkin in a chronic animal model of airway inflammation induced by IFN-gamma. We have revised the Discussion section as (page 13, lines 281 to 286): “First, we utilized an acute mouse model of IFN-gamma treatment to study the role of Parkin in the inflammatory process to mimic the inflammatory process in asthma patients with acute exacerbations which pose a significant healthcare challenge and could contribute to fatal asthma. Future studies are necessary to determine the role of Parkin in chronic type 2-low inflammation induced by IFN-gamma.”

2) Likewise, studying the role of parkin in epithelial cell lines from one human sample may not give adequate insights. Replication of this experiment in epithelial cell lines from other healthy donors would be expected.

For Figure 2 where primary cell cultures were performed, we had cells from n=6 human donors in each group. As we responded to reviewer 1’s comment, the sample size of 6 was based on our initial power calculation.  For Figures 3 and 4, some of samples with n=3 replicates from the same human subject, which is due to the fact that the data were generated from the cell line we created for gene overexpression or gene knockout for the mechanistic studies. Unlike the primary cells from individual subjects, the cell lines with the gene manipulation (overexpression or knockout) are intended for the proof-of-concept studies as most previous studies used the cell line created from a single human donor.  We have acknowledged this in the Discussion section (page 14, lines 297 to 300): “We generated gene overexpression (i.e., Thap11) or knockout (i.e., Parkin) from a single donor to demonstrate their biological functions. Although we expect similar results using cells from different donors with gene overexpression or knockout, this needs to be confirmed in future studies.”

3) A variety of factors, including environmental and genetic factors, microbial pathogens, and upregulation of inflammatory mediators, may be involved in the development of airway inflammation, but the detailed causes remain unknown. Based on these backgrounds, the used experimental model in this study (interferon gamma-induced airway inflammation in mice) should be described in detail in the introduction. The inhibitory effects of other agents on this model should be addressed in the introduction.

With your suggestion, we have expanded the introduction about why we chose to use the interferon gamma-induced airway inflammation model to study type 2 low asthma. It stated (page 3, lines 57 to 64): “This was further supported by a recent study demonstrating the association of increased IFN-g expression in airway samples from a cohort of asthma patients with more severe disease despite corticosteroid use, one of the major anti-inflammatory approaches to treat asthma. While corticosteroids such as dexamethasone have been shown to inhibit IFN-gamma signaling in vitro (PMID: 12707366), it does not inhibit IFN-gamma level in a mouse model of asthma (PMID: 29342458). The fact that severe asthma patients presented higher levels of IFN-gamma even with treatment of high doses of corticosteroids (PMID: 26121748) suggests the urgent need to further understand how IFN-gamma-mediated inflammation is regulated.”   

4) In section 2.2., how were the dose of IFN-gamma (5 ng/ml) and the exposure time (24h) decided by the authors? Authors are advised to address this point and add the answers/proper citations to this section.

In our preliminary cell culture experiment, we performed a time course (24 and 48 hr) and dose response (5 and 10 ng/ml) study of IFN-gamma stimulation in human airway epithelial cells (n=3 subjects). We found consistent decrease of Thap11 by 5 ng/ml of IFN-gamma at 24 hr. We have provided this piece of information in the Methods section (page 6, lines 119 to 128). In most of the previous human lung epithelial cell culture publications, IFN-gamma concentrations ranged from 1 to 10 ng/ml (PMID: 31072323, 11748265, 15297269). IFN-gamma dose (5 mg/ml) used in our study falls within the IFN-gamma dosing range of previous work.

5) In the material and methods section, please comment on the adequacy of sample size calculation. How did the authors decide on using the listed number of animals per experimental group? Please, clarify for readers.

In Figure 1, we have 6 mice for each experimental group. The number of mice for each group was decided based on our previous publication studying the role of Parkin in airway type 2 inflammation (PMID: 32499407) where 5 to 7 mice per experimental group were included. We have provided this piece of information in our revised manuscript (page 22, lines 448 to 451).

6) In Western blotting (section 2.5), the authors should describe in more detail the way Western blotting was conducted. How many micrograms were loaded into the gel, how was the transfer done (wet or semidry), how was band visualization performed; did you use x-ray film or specific imaging equipment? Please, add the answer to the comment to this section.

We have provided more details for western blot in our revised methods section (pages 7 and 8, lines 162 to 170). It stated: “Cell lysates (15 µg of total protein) were separated by SDS-PAGE, transferred onto PVDF membranes using a semidry transfer protocol, blocked with blocking buffer, and incubated with antibodies against Parkin, Thap11, or beat-actin (1:500, Santa Cruz Biotechnology, Dallas, Texas, USA). After washes in PBS with 0.1% Tween-20, the membranes were incubated with the appropriate horseradish peroxidase (HRP)-linked secondary antibodies (1:3000; EMD Millipore, Burlington, Massachusetts, USA) and developed using a Fotodyne imaging system (Fotodyne, Inc., Hartland, Wisconsin, USA). Densitometry was performed to quantify protein expression levels using the National Institutes of Health’s ImageJ software.”

7) In section 2.6, please add the catalog numbers for the used kits.

The catalog numbers for the Human IL-8 (catalog number: DY208) and mouse LIX/CXCL5 (catalog number: DY443) ELISA kits have been added in the Methods section (page 8, lines 173 to 174).

8) In the statistical analysis section, did the authors check data normality before proceeding to one-way ANOVA? Authors are advised to address this point and add the answers to the comment in this section.

We checked data normality before proceeding to one-way ANOVA. We have added the information in the Methods section by stating (page 8, lines 177 to 178): “For parametric data with a normal distribution, a paired Student’s t-test was performed for two-group comparisons or two-way ANOVA followed by the Tukey’s multiple comparison test.”

9) In Figure legends, the authors are advised to clarify in Western blotting whether the data were extracted from independent samples.

We have clarified that in western blotting the data were extracted from independent samples.

10) The full name of abbreviations should be given at the first mention of the abbreviation in the text. For example, BAL in line 85. Please address this issue in the entire manuscript.

We have provided the full name of abbreviations at their first mention.

Reviewer 3 Report

I have no reservations.

Graphical abstract could make this paper even more attractive.

Author Response

A graphical abstract has been provided as Figure 5 in our revised manuscript.

Thank you!

Round 2

Reviewer 2 Report

The authors adequately addressed the raised comments. Thanks!

Minor editing of the English language is required.